# The Relationship between Low-Sodium Salt Intake and Both Blood Pressure Level and Hypertension in Chinese Residents

**DOI:** 10.3390/nu16121909

**Published:** 2024-06-17

**Authors:** Cuicui Wang, Zilong Lu, Jiyu Zhang, Xiaorong Chen, Jianwei Xu, Bingyin Zhang, Jing Dong, Jie Ren, Chunxiao Xu, Congcong Gao, Xiaolei Guo, Jing Wu, Jixiang Ma

**Affiliations:** 1School of Public Health, Cheeloo College of Medicine, Shandong University, Jinan 250100, China; wcc3075284752@163.com; 2The Department for Chronic and Non-Communicable Disease Control and Prevention, Shandong Center for Disease Control and Prevention, Jinan 250014, China; lzllzl22@163.com (Z.L.); zhangjiyu008@163.com (J.Z.); 13153785152@163.com (B.Z.); freedom.dj1@163.com (J.D.); qdcdcrenjie@126.com (J.R.); xuchunxiao919@163.com (C.X.); gaocongcong0110@163.com (C.G.); 3National Center for Chronic and Noncommunicable Disease Control and Prevention, Chinese Center for Disease Control and Prevention, Beijing 100050, China; xiaorch@126.com (X.C.); xujianwei@ncncd.chinacdc.cn (J.X.); 4Shandong Center for Disease Control and Prevention, Jinan 250014, China

**Keywords:** low-sodium salt intake, systolic blood pressure, diastolic blood pressure, hypertension, dose–response

## Abstract

Compared to common salt, low-sodium salt can reduce blood pressure to varying degrees. However, the exact dosage relationship remains unclear. We aimed to investigate the dose–response relationships between low-sodium salt intake and systolic blood pressure (SBP) and diastolic blood pressure (DBP), as well as the risk of hypertension, and to determine the optimal range for low-sodium salt intake. We investigated the basic characteristics and dietary profile of 350 individuals who consumed low-sodium salt. The samples were divided into three groups according to the 33.3rd and 66.6th percentiles of low-sodium salt intake in condiments (Q1: <4.72 g/d, Q2: ≥4.72 g/d, and <6.88 g/d, and Q3: ≥6.88 g/d). The restricted cubic spline results indicated that low-sodium salt intake decreased linearly with SBP and DBP, while low-sodium intake demonstrated a non-linear, L-shaped relationship with the risk of hypertension, with a safe range of 5.81 g to 7.66 g. The multiple linear regression analysis revealed that compared with group Q1, the DBP in group Q2 decreased by 2.843 mmHg (95%CI: −5.552, −0.133), and the SBP in group Q3 decreased by 4.997 mmHg (95%CI: −9.136, −0.858). Exploratory subgroup analyses indicated that low-sodium salt intake had a significant impact on reducing SBP in males, DBP in females, SBP in rural populations, and DBP in urban populations. The intake of low-sodium salt adheres to the principle of moderation, with 5.81–7.66 g potentially serving as a pivotal threshold.

## 1. Introduction

Hypertension is a cardiovascular syndrome with elevated systemic arterial blood pressure as its primary clinical manifestation. According to the World Health Organization (WHO) [1], 1.28 billion adults aged 30 to 79 years worldwide have suffered from hypertension. Furthermore, a study found that high systolic blood pressure can lead to an estimated 10.8 million avoidable deaths every year, and a burden of 235 million years of life lost or lived with a disability (disability-adjusted life years, DALYs) annually [2,3]. In China, there were 245 million hypertensive patients over the age of 18 in 2015 [4,5], and 2.54 million people died of elevated SBP in 2017, with DALYs exceeding 5% [4,6]. Due to population aging and growth, the hypertension prevalence in China continues to increase, posing significant health and economic challenges. Therefore, it is crucial to outline scientific and practical measures for hypertension prevention and control.

High sodium and low potassium dietary intake are well-established risk factors for hypertension and its complications [7]. Therefore, controlling dietary sodium intake is an effective measure to control hypertension [8,9,10,11,12,13]. Using salt substitution or low-sodium salt is an appealing strategy in promoting education and behavior guidance for controlling dietary sodium intake and managing hypertension, given that it reduces sodium intake while increasing potassium intake. Low-sodium salt is a table salt that uses sodium chloride as a carrier and adds a certain amount of magnesium and potassium salts [14]. Compared to regular salt, low-sodium salt contains less sodium and more potassium and magnesium. A cluster-randomized trial [15] conducted in 48 nursing homes by the Chinese scientists Wu Yangfeng et al. showed that, compared to regular salt, salt substitutes reduced systolic blood pressure (SBP) by 7.1 mmHg (95% CI: 3.8–10.5 mmHg) and diastolic blood pressure (DBP) by 1.9 mmHg (95% CI: 0.2–3.6 mmHg) in older adults and reduced the risk of cardiovascular events (hazard ratio: 0.60, 95% CI: 0.38–0.96). Another double-blinded randomized controlled trial [16] designed to test the effects of a salt substitute on 608 participants in northern rural China showed a 3.7 mmHg reduction in SBP levels with the salt substitution strategy. However, the optimal intake level of low-sodium salt remains unknown. Furthermore, there are limited studies on the dose–response relationship between low-sodium salt intake and blood pressure and hypertension. Moreover, there is a lack of data concerning how this relationship varies among different populations.

Therefore, our study used data from the 2016 project evaluation dietary survey of the Shandong-MOH action on salt and hypertension (SMASH) to analyze the relationship between low-sodium salt intake and SBP, DBP, and the risk of hypertension among people with low-sodium salt intake using restricted cubic spline function and multiple linear regression. Exploratory subgroup analyses were carried out in different populations. The dose–response relationships between low-sodium salt intake and blood pressure and hypertension were clarified to provide a scientific basis for lowering blood pressure through low-sodium salt intake.

## 2. Materials and Methods

### 2.1. Data Source

We used data from the 2016 project evaluation dietary survey of the Shandong-MOH action on salt and hypertension (SMASH). Detailed information on the study population is reported in other publications [8]. The data set comprised a total of 1977 individuals, with those who reported having high blood pressure (205 people) initially excluded. This resulted in a sample size of 1772 individuals. Subsequently, those who did not consume low-sodium salt (1422 individuals) were removed, resulting in a final sample size of 350 (Appendix A). All subjects provided informed consent for inclusion before they participated in the study. The study was conducted in accordance with the Declaration of Helsinki, and the protocol was approved by the Preventive Medicine Ethics Committee of Shandong Provincial Centre for Disease Control and Prevention (approval number 2016-7 from 29 April 2016).

### 2.2. Measurement of Low-Sodium Salt Intake

We used dietary investigation methods [17,18] in this study to measure the intake of low-sodium salt. The methodology encompassed a comprehensive recording of all foods consumed by the participants, except for condiments, over a period of three consecutive days, including one weekend day (either Saturday or Sunday). This recording was achieved through a 24-h recall approach. The assessment specifically included processed foods (referring to pre-packaged foods (including pickles) and salt-containing foods (such as dumplings, steamed buns, etc.) made by catering units, excluding condiments). Concurrently, the daily consumption of various condiments in the families of the survey subjects was recorded using a weighing method, and the per capita consumption of condiments of the survey subjects was calculated. The sodium intake of various condiments of the survey subjects was calculated according to the sodium content of various condiments in the Chinese food composition table, and the actual intake of salt of the condiments and the actual intake of low-sodium salt of the condiments were finally obtained according to the molecular weight of sodium chloride and sodium. The diet survey was carried out in the way of on-site entry, and all investigators were uniformly trained by provincial teachers. According to the 33.3rd and 66.6th percentiles of low-sodium salt intake in condiments, the populations were divided into three groups (group Q1: <4.72 g/d, group Q2: ≥4.72 g/d, and <6.88 g/d, and group Q3: ≥6.88 g/d).

### 2.3. Definition of Hypertension

The criteria for diagnosing hypertension [7] in this study were the mean of three measurements of blood pressure without the use of antihypertensive medication: SBP ≥ 140 mmHg and/or DBP ≥ 90 mmHg. Self-reported hypertension was defined according to the data gathered through the study’s individual questionnaires: (1) the use of antihypertensive drugs in the last two weeks or (2) having taken antihypertensive drugs on the day of the measurement. Meeting any of these criteria was considered self-reported hypertension.

### 2.4. Covariates

Study covariates included age; gender; region; marital status; educational level; income; smoking status; drinking status; physical exercise; body mass index (BMI); dyslipidemia [19]; diabetes [20]; and central obesity [21]. Dyslipidemia was defined as one of TC ≥ 6.22 mmol/L or TG ≥ 2.26 mmol/L or LDL-C ≥ 4.14 mmol/L or HDL-C < 1.04 mmol/L. Diabetes mellitus was defined as one of the following conditions: (1) fasting blood glucose ≥ 7.0 mmol/L in the respondent’s laboratory examination results; (2) an oral glucose tolerance test was performed, and the blood glucose was ≥11.1 mmol/L after 2 h of oral glucose intake; and (3) self-reported diabetes mellitus diagnosed by a hospital at or above the provincial level, or by a municipal hospital, county hospital, or township hospital. Central obesity was defined as a waist circumference ≥90 cm in males or ≥85 cm in females. The specific definitions of the variables can be found in Appendix A.

### 2.5. Statistical Analyses

The qualitative data are presented as cases or percentages, and comparisons between the groups were made using the chi-squared test or Fisher’s exact test. Quantitative data are expressed as the mean ± standard deviation, with differences between groups assessed using the T-test or ANOVA. We employed a restricted cubic spline to analyze the dose–response relationships between low-sodium salt intake and blood pressure and the risk of hypertension in the fully-adjusted model. We used three knots, corresponding to the 10, 50, and 90th percentiles. Multiple linear regression was used to analyze the association between low-sodium salt intake and both SBP and DBP. This analysis adjusted for several potential confounding factors, including age, gender, geographic region, marital status, education level, income, smoking and drinking habits, physical activity, BMI, dyslipidemia, diabetes, and central obesity. Low-sodium salt intake was modeled both as a categorical and a continuous variable. In order to examine the association between low-sodium salt intake and outcomes in populations with different demographic characteristics, exploratory subgroup analyses were conducted by sex (male and female), age group (17–44 and 45–70 years), and region of residence (urban and rural). Multiplicative interaction analyses were used to examine the relationship between low-sodium intake and the subgroup factors. Additionally, to account for the effect of other salt consumption on low-sodium salt intake, another type of salt intake (i.e., the actual intake of salt in condiments minus the actual intake of low-sodium salt in condiments) was included in the model as a covariate for sensitivity analysis. All statistical tests were two-sided, and *p* < 0.05 was considered statistically significant. The data analyses were performed using R software version 4.2.2.

## 3. Results

### 3.1. Baseline Characteristics of the Study Populations

A total of 1772 people were divided into two groups according to whether they consumed low-sodium salt or not. As demonstrated in Appendix A, the differences between the basic characteristics of the two groups, such as age (41.97 vs. 40.85, *p* = 0.136) and gender (*p* = 0.676), were not statistically significant, with the exception of the region. This study included 350 participants, with an average age of 40.85 years (standard deviation, 12.49 years), of which 51.1% were male. Compared with those in group Q1, those in group Q3 had a higher salt intake and lower prevalence of hypertension. In group Q3, the proportion of males was higher than that of females, and the current non-smoker group exceeded that of the current smoker group. These details are presented in Table 1.

### 3.2. The Relationship between Low-Sodium Salt Intake and Both SBP and DBP

In the restricted cubic spline regression models, the correlations between low-sodium salt intake and SBP and DBP linearly decreased, as shown in Figure 1a,b (*p* < 0.0001). As shown in Table 2, after adjusting for age, gender, region, marital status, education level, income, smoking status, drinking status, physical exercise, BMI, dyslipidemia, diabetes, and central obesity (Model 4), an increase in low-sodium salt intake resulted in a significant reduction in DBP in group Q2 by 2.843 mmHg (β = −2.843; 95% CI: −5.552, −0.133) relative to group Q1. Similarly, compared to group Q1, the SBP in group Q3 decreased by 4.997 mmHg (β = −4.997; 95% CI: −9.136, −0.858).

### 3.3. The Relationship between Low-Sodium Salt Intake and Hypertension

In the restricted cubic spline regression model, the correlation between low-sodium salt intake and the risk of hypertension was non-linear, as shown in Figure 1c (*p* = 0.0118). Specifically, low-sodium salt intake was a protective factor when it ranged from 5.81 g to 7.66 g; however, it was a risk factor when it was less than this range, and the results were not statistically significant when it was higher than this range.

### 3.4. Exploratory Subgroup Analyses

Exploratory subgroup analyses across various cohorts were undertaken to explore the potential effect modification on the associations between low-sodium salt intake and SBP, DBP, and the risk of hypertension. As shown in Appendix A, the restricted cubic spline results were consistent with the overall population results. In different population subgroups, SBP and DBP showed a decreasing trend with increasing low-sodium salt intake. Notably, in the rural subgroup, the relationship between low-sodium salt intake and hypertension risk exhibited a non-linear pattern. The relationship showed a tendency to decrease and then increase. Specifically, low-sodium salt intake was a protective factor when it was between 5.98 g and 6.28 g; however, it was a risk factor when it was less than this range, and the results were not statistically significant when it was higher than this range. As shown in Appendix A, after adjustment for all potential covariates, with increasing low-sodium salt intake, compared to group Q1, SBP in group Q3 decreased by 6.435 mmHg (β = −6.435; 95% CI: −11.708, −1.161) in males, DBP in group Q2 decreased by 4.082 mmHg (β = −4.082; 95% CI: −8.141, −0.022) in females, SBP in group Q3 decreased by 6.950 mmHg (β = −6.950; 95% CI: −12.808, −1.092) in rural areas, and DBP in group Q2 decreased by 4.477 mmHg (β = −4.477; 95% CI: −8.873, −0.081) in urban residents. The results of the multiplicative interaction analyses indicated that there were no interactions between low-sodium intake and gender, age group, or region (*p* for interaction > 0.05).

### 3.5. Sensitivity Analysis

Considering the effect of other salt consumption on low-sodium salt intake, we incorporated the intake of other salt consumption as a covariate into the model.

The statistical analysis results showed that compared with group Q1, the DBP in group Q2 decreased by 2.845 mmHg (β = −2.845; 95% CI: −5.559, −0.131), and the SBP in group Q3 decreased by 4.961 mmHg (β = −4.961; 95% CI: −9.142, −0.779) (Appendix A). Sensitivity analyses were also performed in different subgroup populations; after adjusting for all covariates, among females, the change in DBP was not statistically significant in group Q2 compared to group Q1. The rest of the results were similar to those of the main body model (Appendix A).

## 4. Discussion

In this cross-sectional study of Shandong Province residents in China, we observed that higher low-sodium salt intake was associated with lower SBP and DBP. Additionally, DBP is more sensitive to low-sodium salt intake than SBP. As low-sodium salt intake increased, a decrease in DBP occurred in group Q2 (the intake of low-sodium salt was ≥4.72 g/d and <6.88 g/d); however, a decrease in SBP did not occur until group Q3 (the intake of low-sodium salt was ≥6.88 g/d). Moreover, the decrease in SBP was greater than that in DBP. Notably, the relationship between low-sodium salt intake and the risk of hypertension was non-linear, characterized by a decline followed by a rise, presenting an L shape with a safe range of 5.81 g–7.66 g. Furthermore, exploratory subgroup analyses indicated that low-sodium salt intake had a significant impact on reducing SBP in males, DBP in females, SBP in rural populations, and DBP in urban populations. Specifically, in the rural population, the relationship between low-sodium salt intake and the risk of hypertension was non-linear, showing a decrease followed by an increase, presenting an L shape with a safe range of 5.98 g–6.28 g.

Many studies have now demonstrated that reducing sodium intake and/or increasing potassium intake can lower blood pressure [22,23,24,25]. Additionally, a range of domestic and international randomized controlled trials have demonstrated that individuals who consume low-sodium salt or salt substitutes exhibit varying degrees of reductions in SBP and DBP compared to those who consume regular salt [26,27,28,29,30]. These findings offer indirect evidence supporting the conclusion that the intake of low-sodium salt is associated with reductions in both SBP and DBP. In our dose–response analysis, we identified a linear relationship between the consumption of low-sodium salt and reductions in both SBP and DBP. The tertile analysis revealed that, compared to the lowest group, individuals in the highest group of low-sodium salt intake experienced an average SBP reduction of 4.997 mmHg (β = −4.997; 95% CI: −9.136, −0.858). However, the intake levels in the middle group did not significantly impact SBP reduction. This phenomenon may be attributed to the intake in the highest group reaching a threshold that affects salt sensitivity due to changes in the potassium and sodium levels, which in turn influences the SBP [31]. While the effects on the DBP varied, an increase in low-sodium salt intake from the lowest to the middle group resulted in a significant reduction in DBP by 2.843 mmHg (β = −2.843; 95% CI: −5.552, −0.133). This notable decrease can be attributed to DBP’s primary sensitivity to peripheral vascular resistance, especially in the small arteries and capillaries [32]. Low-sodium salt intake initially reduces resistance in these peripheral vessels, thereby influencing the velocity and pressure of the blood flowing back to the heart, making DBP more responsive to changes in sodium intake [32,33]. Moreover, while the upper group analysis initially showed statistically significant effects before adjustments for dyslipidemia, diabetes, and central obesity—aligning with other studies in demonstrating a smaller reduction compared to SBP [26,30,34,35,36]—the significance dissipated after these adjustments. This could be due to the influence of these health conditions or perhaps the small sample size of the study, which could lead to inconsistent outcomes, thus highlighting the need for further investigation. From another perspective, when looking at the results of the fully-adjusted model 4, compared to group Q1, there was a meaningful decrease in DBP in group Q2, whereas there was no significance in group Q3, suggesting that a non-linear relationship may be more applicable to the association between low-sodium salt intake and DBP than a linear one. It also demonstrates the need for further analysis. Furthermore, we conducted exploratory analyses in different gender populations. The results of our exploratory subgroup analyses indicated that low-sodium salt intake effectively reduces SBP in males and DBP in females, suggesting a gender difference in the benefits of low-sodium salt intake on blood pressure. Hu Jihong’s study [37] also confirmed that the blood pressure-lowering effects of low-sodium salt vary between genders; those with higher initial blood pressure and women experienced more significant reductions in household blood pressure within the salt substitute group compared to those using regular salt. However, a study on the effect of low-sodium salt substitutes on the blood pressure of a rural population in North China [38] found that low-salt substitutes (65% NaCl, 25% KCl, 10% MgSO4) had similar beneficial effects on both men and women. Furthermore, a meta-analysis [13] showed a significant reduction in SBP for both genders. Therefore, the differences observed, potentially due to varying hormone levels among the different genders or possibly due to our smaller sample size, may have resulted in discrepancies with other studies and necessitate further research.

Previous studies [35] have shown that a modest reduction in salt intake by 2 g over 18 months can lead to a 35% decrease in the incidence of hypertension over a 7-year follow-up period among individuals with initially normal blood pressure. Our study found a non-linear correlation between low-sodium salt intake and the risk of hypertension in populations that consume low-sodium salt. This correlation was also observed in the rural population subgroup analysis. With the increase in low-sodium salt consumption, the risk of hypertension showed a trend of decreasing and then increasing. This may be because the intake of low-sodium salt has a lowering effect on blood pressure, but the excessive intake of low-sodium salt leads to a higher total salt intake, which leads to a corresponding increase in the risk of hypertension. Hence, it becomes clear that there is a threshold to the beneficial effects of low-sodium salt intake, identified to be around 5.81 g–7.66 g, guiding the optimal consumption level of low-sodium salt to avoid excessive intake. At the same time, we observed that when the consumption of low-sodium salt exceeded 7.66 g, its relationship with the risk of hypertension ceased to be statistically significant. This phenomenon may be attributed to the increased potassium intake associated with a higher consumption of low-sodium salt, which facilitates sodium excretion, thus mitigating the adverse effects of increased sodium intake. Additionally, the small sample size of our study may have influenced these findings, underscoring the need for further research. Differences between urban and rural populations may be due to urban residents having dietary habits that include higher consumption of fruits and vegetables compared to rural areas, resulting in higher dietary potassium intake, which in turn may diminish the effect of low-sodium salt among urban residents [39,40]. However, our study lacks the requisite data to confirm these hypotheses, and more comprehensive investigations are required.

Our research pioneers the examination of the association between low-sodium salt consumption and blood pressure, exploring the possible dose–response relationship in the Shandong Province based on the reliable data source known as SMASH. Importantly, it establishes the consumption thresholds for low-sodium salt. This contributes to a critical scientific foundation for employing low-sodium salt in hypertension management and prevention strategies. To some extent, our results make up for the shortcomings of previous studies. However, our study still has several limitations. Firstly, the sample size was relatively small, which may have limited the statistical power and accuracy of the findings. Secondly, due to its cross-sectional design, our study was not equipped to establish causality. Finally, the study only included residents of the Shandong Province who consumed low-sodium salt, which may have limited the generalizability of the findings to the wider population. To conduct a more comprehensive analysis in the future, a larger sample size is needed.

## 5. Conclusions

With the increase in low-sodium salt intake, both SBP and DBP showed a decreasing trend, whereas the risk of hypertension first decreased and then increased. DBP is more sensitive to low-sodium salt intake than SBP. As low-sodium salt intake increased, a decrease in DBP occurred in group Q2 (the intake of low-sodium salt was ≥4.72 g/d, and <6.88 g/d); however, a decrease in SBP did not occur until group Q3 (the intake of low-sodium salt was ≥6.88 g/d). Moreover, the decrease in SBP was greater than that in DBP. However, the intake of low-sodium salt adheres to the principle of moderation, with 5.81 g–7.66 g potentially serving as a pivotal threshold beyond which the benefits may not continue to accrue. Additionally, the relationship between low-sodium salt intake and health outcomes may differ among various populations, underscoring the importance of customized dietary guidelines to address the distinct needs of diverse groups.

## Figures and Tables

**Figure 1 nutrients-16-01909-f001:**
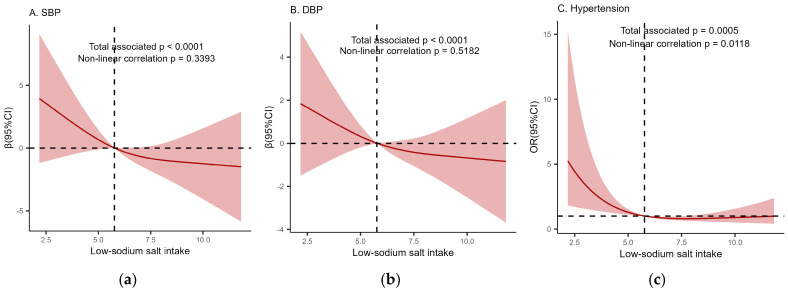
(**a**) The relationship between low−sodium salt intake and SBP; (**b**) the relationship between low−sodium salt intake and DBP; (**c**) the relationship between low−sodium salt intake and the risk of hypertension. The restricted cubic spline model of the relationship model was adjusted for age, gender, region, marital status, education level, income, smoking status, drinking status, physical exercise, BMI, dyslipidemia, diabetes, and central obesity. The horizontal dashed line represents the case where β = 0 or OR = 1, while the vertical dashed line indicates the vertical line through the point (the point where the red solid line intersects the horizontal dashed line).

**Table 1 nutrients-16-01909-t001:** Characteristics of the 350 study participants according to low-sodium salt intake.

Characteristics	Low-Sodium Salt Intake				*p*-Value ^1^
	Total Cases	Q1 (<4.72 g/d)	Q2 (4.72–6.88 g/d)	Q3 (≥6.88 g/d)	
Number of participants, *N*	350	116	114	120	
Salt intake (mean (SD))	8.67 (4.24)	5.62 (2.42)	8.13 (2.33)	12.15 (4.51)	**<0.001**
Low-sodium salt intake (mean (SD))	6.16 (2.92)	3.47 (0.83)	5.77 (0.61)	9.14 (2.79)	**<0.001**
SBP (mean (SD))	119.76 (17.77)	121.99 (22.44)	120.01 (17.04)	117.37 (12.38)	0.134
DBP (mean (SD))	76.43 (11.76)	78.34 (14.17)	76.07 (11.62)	74.93 (8.82)	0.078
Hypertension, N%					
No	302 (86.3)	92 (79.3)	97 (85.1)	113 (94.2)	**0.004**
Yes	48 (13.7)	24 (20.7)	17 (14.9)	7 (5.8)	
Age (mean (SD))	40.85 (12.49)	40.80 (11.69)	41.24 (12.65)	40.52 (13.15)	0.907
Age groups, N%					
17–44 years old	206 (58.9)	68 (58.6)	64 (56.1)	74 (61.7)	0.690
45–70 years old	144 (41.1)	48 (41.4)	50 (43.9)	46 (38.3)	
Gender, N%					
Female	171 (48.9)	68 (58.6)	55 (48.2)	48 (40.0)	**0.016**
Male	179 (51.1)	48 (41.4)	59 (51.8)	72 (60.0)	
Region, N%					
Urban	133 (38.0)	49 (42.2)	44 (38.6)	40 (33.3)	0.366
Rural	217 (62.0)	67 (57.8)	70 (61.4)	80 (66.7)	
Marital status, N%					
Unmarried ^2^	42 (12.0)	16 (13.8)	9 (7.9)	17 (14.2)	0.258
Married	308 (88.0)	100 (86.2)	105 (92.1)	103 (85.8)	
Educational level, N%					
Primary and below	105 (30.0)	32 (27.6)	33 (28.9)	40 (33.3)	0.829
Junior high school	153 (43.7)	51 (44.0)	53 (46.5)	49 (40.8)	
High school and above	92 (26.3)	33 (28.4)	28 (24.6)	31 (25.8)	
Income ^3^, N%					
≥¥0, ≤¥5000	102 (30.0)	30 (26.3)	38 (34.5)	34 (29.3)	0.678
>¥5000, ≤¥10,000	101 (29.7)	35 (30.7)	29 (26.4)	37 (31.9)	
>¥10,000, ≤¥15,000	59 (17.4)	19 (16.7)	17 (15.5)	23 (19.8)	
>¥15,000	78 (22.9)	30 (26.3)	26 (23.6)	22 (19.0)	
BMI (mean (SD))	24.76 (4.08)	25.32 (4.10)	24.88 (4.43)	24.11 (3.63)	0.071
BMI groups, N%					
<24 kg/m^2^	156 (44.6)	48 (41.4)	50 (43.9)	58 (48.3)	0.407
≥24 kg/m^2^, <28 kg/m^2^	128 (36.6)	40 (34.5)	43 (37.7)	45 (37.5)	
≥28 kg/m^2^	66 (18.9)	28 (24.1)	21 (18.4)	17 (14.2)	
Smoking status, N%					
No	252 (72.0)	92 (79.3)	85 (74.6)	75 (62.5)	**0.012**
Yes	98 (28.0)	24 (20.7)	29 (25.4)	45 (37.5)	
Drinking status, N%					
No	217 (62.0)	82 (70.7)	66 (57.9)	69 (57.5)	0.062
Yes	133 (38.0)	34 (29.3)	48 (42.1)	51 (42.5)	
Physical exercise, N%					
No	243 (69.4)	75 (64.7)	88 (77.2)	80 (66.7)	0.086
Yes	107 (30.6)	41 (35.3)	26 (22.8)	40 (33.3)	
Dyslipidemia, N%					
No	239 (68.3)	77 (66.4)	73 (64.0)	89 (74.2)	0.216
Yes	111 (31.7)	39 (33.6)	41 (36.0)	31 (25.8)	
Diabetes, N%					
No	331 (94.6)	110 (94.8)	106 (93.0)	115 (95.8)	0.623
Yes	19 (5.4)	6 (5.2)	8 (7.0)	5 (4.2)	
Center obesity, N%					
No	203 (58.0)	60 (51.7)	67 (58.8)	76 (63.3)	0.192
Yes	147 (42.0)	56 (48.3)	47 (41.2)	44 (36.7)	

The values are presented as the mean (SD) for the continuous variables or a number (percentage) for the categorical variables. Abbreviations: SD, standard deviation; SBP, systolic blood pressure; DBP, diastolic blood pressure; BMI, body mass index; ^1^ Obtained using the chi-squared test for the categorical variables and one-factor ANOVA for the continuous variables. ^2^ Unmarried means single, divorced, separated, or widowed. ^3^ Income had 10 missing values. The bold values are statistically significant.

**Table 2 nutrients-16-01909-t002:** Relationships between low-sodium salt intake and SBP and DBP.

Outcome	Low-Sodium Salt				*p*-Value for 1 g Increment
	Q1	Q2	Q3	Per 1 g Increment	
SBP					
Model 1	Ref	−1.980 (−6.576, 2.616)	**−4.616 (−9.154, −0.079)**	−0.517 (−1.157, 0.123)	0.113
Model 2	Ref	−3.476 (−7.941, 0.989)	**−6.884 (−11.329, −2.439)**	**−0.739 (−1.359, −0.120)**	**0.020**
Model 3	Ref	−2.829 (−7.036, 1.379)	**−5.051 (−9.244, −0.858)**	−0.488 (−1.072, 0.096)	0.101
Model 4	Ref	−3.112 (−7.282, 1.058)	**−4.997 (−9.136, −0.858)**	−0.460 (−1.037, 0.116)	0.117
DBP					
Model 1	Ref	−2.263 (−5.301, 0.775)	**−3.406 (−6.405, −0.407)**	−0.377 (−0.801, 0.046)	0.081
Model 2	Ref	**−3.124 (−6.166, −0.083)**	**−4.412 (−7.439, −1.384)**	**−0.472 (−0.894, −0.050)**	**0.028**
Model 3	Ref	**−2.755 (−5.485, −0.025)**	**−2.755 (−5.476, −0.035)**	−0.252 (−0.631, 0.127)	0.192
Model 4	Ref	**−2.843 (−5.552, −0.133)**	−2.683 (−5.372, 0.006)	−0.233 (−0.608, 0.142)	0.222

Model 1: not adjusted. Model 2: adjusted for age, gender, region, marital status, education level, and income. Model 3: adjusted for model 2 and smoking status, drinking status, physical exercise, and BMI. Model 4: adjusted for model 3 and dyslipidemia, diabetes, and central obesity. The bold values are statistically significant.

## Data Availability

The original contributions presented in the study are included in the article and Appendix A; further inquiries can be directed to the corresponding authors.

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
