# Peer review of "The Relationship between Low-Sodium Salt Intake and Both Blood Pressure Level and Hypertension in Chinese Residents"

_nutrients, 2024, doi:10.3390/nu16121909_

Round 1

Reviewer 1 Report

Comments and Suggestions for Authors

Dr. Wang & colleagues set up a cross-sectional study with the aim of investigating the dose-response relationship between "low-sodium salt intake" and both blood pressure and the risk of hypertension. They also wanted to assess the optimal range for "low sodium salt” intake. The authors investigated the basic characteristics and dietary profile of 350 individuals (taken from the 2016 project evaluation dietary survey of the Shandong-MOH

action on salt and hypertension [SMASH]) who consumed low-sodium salt divided in tertiles. They found that "low-sodium salt intake" was linearly associated with SBP and DBP and non-linearly with the risk of hypertension. After further stratification, the authors found that different genders and rural vs urban populations apparently responded the most in SBP or DBP. Finally, the authors propose that "the principle of moderation, with 5.81g-7.66g, may serve as a pivotal threshold."

The article is interesting, but I have several concerns.

First, it is unclear how the 350 individuals were chosen from the SMAH study: either for their low salt intake or because they represent the whole sample or whatever? A table should present the different characteristics between these individuals and the entire cohort.

Then, no data about responses to the survey were given, mining the sample's representativeness.

In my opinion, the term "Low-sodium Salt Intake" can be misleading because when you think about salt, you think about sodium, but the authors here mean salt constituted by other minerals, not sodium. Maybe the authors can endeavor to explain the term better by adding, at least at the beginning, the specification that the salt is rich in magnesium and potassium.

Also, the term "the three-quarter percentile" is difficult to understand. I think the authors should simply say that they divided the samples into tertiles according to the 33.3th and 66.6th percentiles.

The legend of Table 2 does not clearly indicate that bold values are statistically significant. Moreover, if you look at average values for DBP in the fully adjusted model (4), they are more compatible with a threshold effect than a linear one. This should be mentioned and commented on.

The study is underpowered to make meaningful stratified analyses. This should be discussed, and the analysis should be presented as exploratory.

Comments on the Quality of English Language

English Language quite well.

Reviewer 2 Report

Comments and Suggestions for Authors

A good paper, using a quite interesting and novel method of statistical approach. However, in the methods chapter, maybe a more detailed description of the restricted cubic spline model would have been of use, as it is quite complicated and a description of how the number and position of the knots were established, and also which was the degree of the polynomials used among the restrictions.

However, as the results were very well presented and the collection of data and their subsequent treatment was irreproachable, there is no reason not to agree that the method used was a novel and efficient one.

The results are the expected ones, there are no breakthroughs in the data presented, so, perhaps a comparison of the method employed, in a few lines, with more traditional statistical methods could have been useful for the reader, in order to stimulate more  researchers to use it.
